# Genetic variation in chromatin state across multiple tissues in *Drosophila melanogaster*

**Khoi Huynh[1], Brittny R. Smith[2], Stuart J. Macdonald[2,3], Anthony D. Long[1]***

**1** Department of Ecology and Evolutionary Biology, University of California, Irvine, California, United States of America, **2** Department of Molecular Biosciences, University of Kansas, Lawrence, Kansas, United States of America, **3** Center for Computational Biology, University of Kansas, Lawrence, Kansas, United States of America

* tdlong@uci.edu

## Abstract

We use ATAC-seq to examine chromatin accessibility for four different tissues in *Drosophila melanogaster*: adult female brain, ovaries, and both wing and eye-antennal imaginal discs from males. Each tissue is assayed in eight different inbred strain genetic backgrounds, seven associated with a reference quality genome assembly. We develop a method for the quantile normalization of ATAC-seq fragments and test for differences in coverage among genotypes, tissues, and their interaction at 44099 peaks throughout the euchromatic genome. For the strains with reference quality genome assemblies, we correct ATAC-seq profiles for read mis-mapping due to nearby polymorphic structural variants (SVs). Comparing coverage among genotypes without accounting for SVs results in a highly elevated rate (55%) of identifying false positive differences in chromatin state between genotypes. After SV correction, we identify 1050, 30383, and 4508 regions whose peak heights are polymorphic among genotypes, among tissues, or exhibit genotype-by-tissue interactions, respectively. Finally, we identify 3988 candidate causative variants that explain at least 80% of the variance in chromatin state at nearby ATAC-seq peaks.

## Author summary

Chromatin states are well described in *Drosophila melanogaster* embryos, but adult and pre-adult tissues are poorly studied, as are differences among genotypes. We carried out ATAC-seq on four different tissues in eight different inbred genotypes with biological replicates within tissue and genotype. We discover that apparent differences in coverage, and by inference chromatin openness, are often due to segregating structural variants (SVs) that can only be corrected for if strains are associated with high-quality genome assemblies. After correction for false positives associated with SVs, we identify thousands of regions that appear to vary in chromatin state between genotypes or vary between genotypes in a tissue-dependent manner. It has been widely speculated that *cis*-regulatory variants contribute to standing variation in complex traits. If this is true, chromatin states that vary between individuals, perhaps in a tissue-dependent manner, are likely to be enriched for quantitative trait loci.

**Data Availability Statement:** The raw fastq files are submitted to NCBI as BioProject: PRJNA761571. A github containing the codes used in this work is here: https://github.com/Kh0iHuynh/ATAC-seq-Project. Several intermediate data tables

resulting from different analyses are hosted here: https://wfitch.bio.uci.edu/~tdlong/sandvox/ publications.html or https://doi.org/10.7280/ D1FM5F. Many of the results, such as ATAC-seq coverage tracks, SNPs/SVs tracks, can also be visualized as Santa Cruz Genome Browser (SCGB) tracks here: http://goo.gl/LLpoNH.

**Funding:** This work was supported by NIH R01 OD010974 (to SJM and ADL), NIH R01 ES029922 (to SJM), and NIH R01 GM115562 (to ADL). We also thank the University of Kansas Genome Sequencing Core facility (funded by NIH P20 GM103638) for assistance with library construction and QC, and the Kansas INBRE program (funded via NIH P20 GM103418) for computational support. The funders had no role in study design, data collection and analysis, decision to publish, or preparation of the manuscript.

**Competing interests:** The authors have declared that no competing interests exist.

## Introduction

Many human complex diseases, such as heart disease and diabetes, are highly heritable [1]. Large high-powered Genome-Wide Association Studies (GWAS) have dominated the study of such diseases over the last decade, but despite thousands of associations between markers and traits, the exact causative variants underlying risk typically remain hidden [2,3], and an appreciable fraction of heritable variation remains unexplained [4]. Recent papers propose that variation in human complex traits is due to thousands of mostly intermediate-frequency, tiny effect variants [5–7]. In contrast, QTL mapping studies in yeast [8,9], mouse [10–12], and *Drosophila* [13] consistently map factors of much larger effect, with mapped QTL collectively explaining a considerable fraction of heritability in a cross. Efforts to fully characterize complex trait loci in model systems may hold the most promise for "lifting the statistical fog" [14] associated with genetic mapping, and point to causative, functional alleles.

A promising strategy for identifying causative variants at candidate genes identified via GWAS or QTL mapping is to focus on regions near those genes that act as *cis*-regulators of gene expression. There is now a preponderance of evidence that the bulk of variation in complex traits is due to regulatory variants [6,15–18], with little evidence that amino acid variants explain human GWAS hits [19]. Yet, so little is actually known about complex traits that even this claim is debated [19,20]. Until recently, non-coding regions with *cis*-regulatory function have been difficult to identify at scale, but genomewide profiling of open chromatin regions using DNase-I HS (DNase-I hypersensitive sites) sequencing [21] and/or the more experimentally straightforward ATAC-seq (Assay for Transposase Accessible Chromatin) approach [22] have allowed characterization of chromatin state in large panels of genotypes [23,24]. ATAC-seq employs the Tn5 transposase sequencing chemistry to make an Illumina-compatible paired-end sequencing library using nucleosome-bound DNA as template for the transposition reaction. Regions of DNA bound by transcription factors or nucleosomes are protected from Tn5 insertion, whereas more open chromatin regions—likely harboring active *cis*-regulatory features—are associated with higher levels of sequence coverage. Much like RNA-seq data, open chromatin regions identified by ATAC-seq can vary among tissues, developmental timepoints, and genotypes [23,25–27]. In terms of the genetics of complex traits, chromatin features displaying variation among genotypes, especially in a tissue-specific manner, are of considerable interest as potential contributors to trait variation.

Multiple DNase1-HS-seq and ATAC-seq studies have been carried out in *Drosophila melanogaster* [26,28–41] as well as other insects such as *Anopheles gambiae* [42]. The majority of *Drosophila* studies have focused on a single genotype (or cell line), have compared different mutant backgrounds, or have employed a small number of wildtype strains that lack a high-quality genome sequence (*c.f.* [26,29,33,35,36,38,39]). In no case has the genotype queried been the *Drosophila melanogaster* reference strain (*i.e.*, Bloomington stock 2057 or "iso1"), the strain ATAC-seq reads are generally aligned to. There are routinely a considerable number of SNPs, short insertion/deletion variants, and a wide array of structural variants (SVs) distinguishing any pair of *Drosophila* strains [43], and such events–if they are effectively "hidden" due to the absence of high-quality genomes for the target strains–may complicate the analysis of chromatin state, as has been observed with RNAseq data [44]. Furthermore, chromatin accessibility studies in *Drosophila* have focused principally on early embryonic stages [26,30–32,34–37,41], cell lines [28], or whole adults, with only five studies examining specific adult tissues or imaginal discs [29,33,38–40]. In terms of the complex traits that tend to be studied in the *Drosophila* research community, which are skewed towards traits measured in adults and larvae (*c.f.* Table 3 of [45]), *cis*-regulatory elements active in imaginal discs or adult tissues are likely of broad interest.

Here we carry out a biologically-replicated ATAC-seq experiment to characterize chromatin accessibility in four adult tissues in several highly-characterized isogenic genotypes of *D. melanogaster* [43] (throughout this paper we use genotype to refer to a genome wide genotype or isogenic strain). We identify a set of peaks with evidence for an open chromatin configuration in at least one of the tissues. Unlike previous studies, and inspired by the quantile normalization method deployed in microarray research [46], we develop a method for normalizing ATAC-seq reads across tissues, genotypes, and biological replicates. We carry out statistical tests to identify ATAC-seq peaks that differ in coverage as a function of tissue, genotype, or that display a tissue-by-genotype interaction. By virtue of studying highly isogenic genotypes with reference quality *de novo* assemblies, we correct for artifacts in peak coverage due to hidden SVs. We show that a failure to correct for the impact of SVs can result in a high rate of peaks inferred as differing between genotypes, which are in fact due to mis-mapped read pairs. We finally identify a set of SNPs near to variable ATAC-seq peaks that potentially represent candidate causal *cis*-acting factors.

## Results

### Workflow and samples

We dissected wing and eye-antennal imaginal discs from male third instar larvae, and brains and ovaries from adult females, for eight *Drosophila* Synthetic Population Resource [47] founder strains. All eight have been re-sequenced using short-read sequencing [48], while seven have extremely well characterized, reference-quality genomes [43]. For each tissue and genotype combination we obtained three biological replicates. The eight genotypes chosen are highly inbred and represent a world-wide sampling of variation within the species (see S1 Fig and S1 Table). Dissected samples were immediately processed to make indexed ATAC-seq libraries [22] and sequenced to obtain 20–147 million Illumina paired-end reads per sample (mean = 73M, SD = 21M). Reads were aligned to the *D. melanogaster* reference genome (dm6) and pooled across genotypes, but within tissues, to identify open chromatin "peaks" located throughout the euchromatic genome using MACS2 [49]. Individual replicate/genotype/tissue samples were separately normalized to obtain a weighted coverage at each identified peak. Finally, we utilized reference quality assemblies for the seven assembled strains to correct read coverage statistics for the presence of nearby polymorphic structural variants (SVs) and carried out statistical tests at peaks to identify chromatin structures that varied among the four tissues, the seven genotypes, or exhibited a tissue-by-genotype interaction. Our general workflow is depicted in S2 Fig and read mapping statistics for each sample are given in S4 Table.

### ATAC-seq identifies open chromatin regions across four tissues in the *Drosophila* genome

Peaks were filtered to only include those in euchromatin regions (see S2 Table) that were also significantly enriched above background at $p<0.01$ as defined by MACS2. After filtering, we identified 25464, 18111, 18496, and 17413 euchromatic peaks for adult female brain, ovary, eye-antennal imaginal disc and wing imaginal disc tissue, respectively. Venn diagrams showing peaks shared among tissues for the set of peaks enriched at $p<0.01$ and at $p<0.001$ (Fig 1A) are qualitatively similar, supporting the idea that the significance threshold for enrichment that is employed only subtly impacts the collection of peaks we consider. The Venn diagram at $p<0.01$ (Fig 1A) shows that although peaks shared among tissues are not uncommon– 9.8%, 7.5%, and 17.2% of the total collection of peaks are shared by all four, three, or two tissues, respectively– 65.6% of the peaks are private to a single tissue. Brain tissue exhibits the highest number of private ATAC-seq peaks, but even the pair of disc tissues–which one might naively

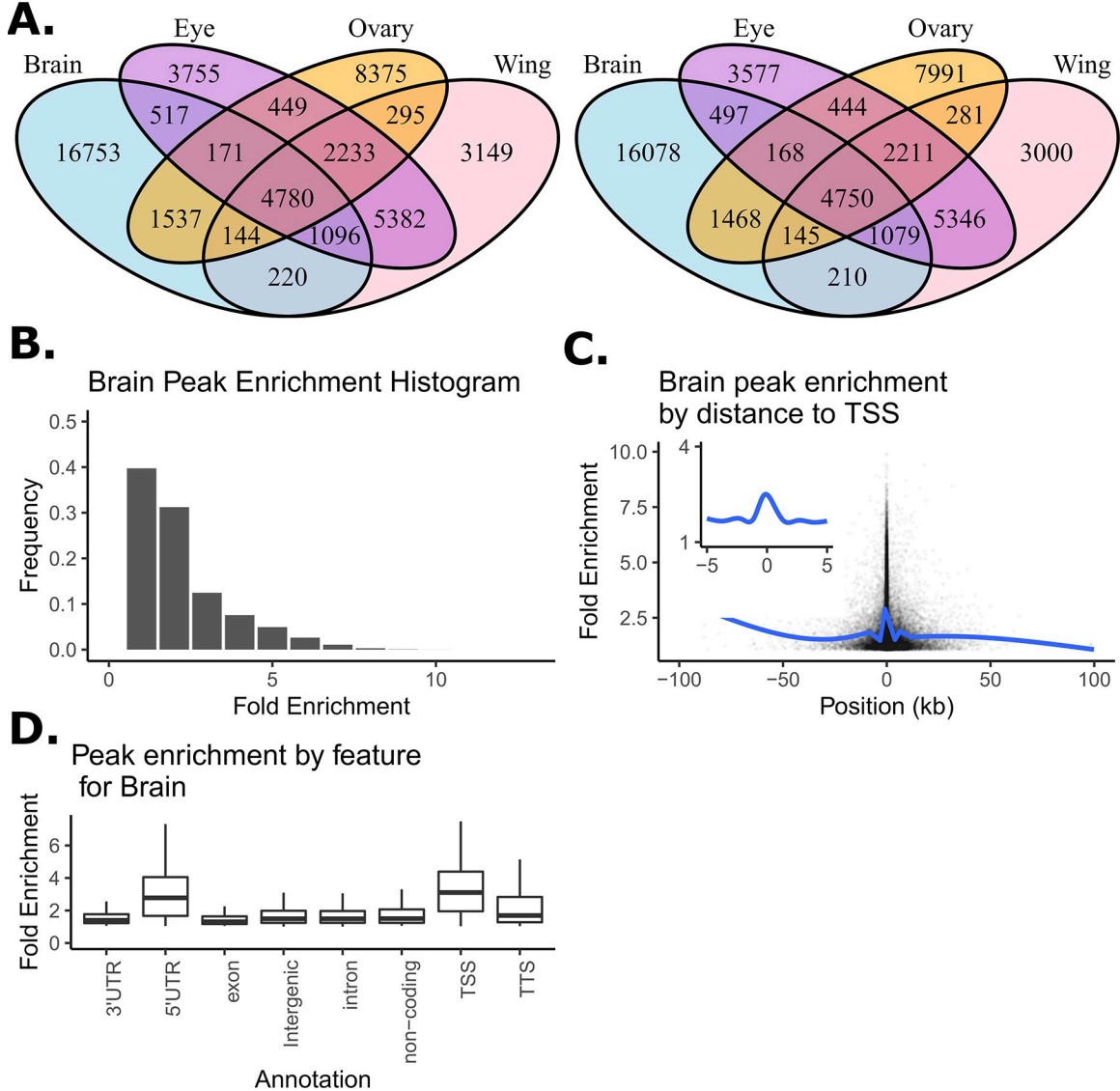

**Fig 1. Summary of open chromatin peaks identified across four tissues.** (A) Venn diagram showing overlap in peak calls across tissues as a function of the p-value cut-offs of 0.01 (left) and 0.001 (right). (B) Distribution of peak enrichment scores for the brain samples. (C) Peak enrichment scores as a function of distance to the nearest transcription start site with a smoothing line for brain samples. Insert focuses on peaks within 10kb of the TSS showing only the smoothing line. (D) Peak enrichment distribution as a function of genomic feature for brain samples (TSS, transcription start site; TTS, transcription termination site).

think would be the most similar of our target tissue types–have appreciable numbers of private peaks, highlighting the value of tissue-specific chromatin characterization.

For each peak, within each tissue, we characterized fold enrichment (a measure of peak "height" based on read count in the peak relative to the local background [49]) to explore whether the properties of the peaks we identify resemble those observed in previous studies. Fig 1B, depicting fold enrichment for the brain, shows that the vast majority of peaks (>90%) have fold enrichments of less than 5. We observed the same trends for the four other tissues (see S3A, S4A and S5A Figs). This observation is consistent with results from DNase1-HS-seq experiments [50,51] and other ATAC-seq datasets [52,53]. We further examined the distribution of fold enrichment as a function of distance from transcription start sites (TSSs) for brain

peaks, as TSSs often exhibit strong enrichment patterns [54,55]. Fig 1C shows fold-enrichment as a function of distance from the TSS for the female brain (S3B, S4B and S5B Figs for the other tissues). The patterns we see largely mirror other studies [52,54,55]. We finally examined average fold enrichment as a function of HOMER annotation type (Fig 1D depicts female brain). Fold enrichments are strongest for 5'UTR, TSS, and perhaps transcription termination sites (TTS). Enrichments are more subtle for other feature types, although for all feature types there were clearly a subset of peaks with strong fold enrichment scores. The same trend in peak enrichment with regard to feature types can also be observed in other tissues (S3C, S4C and S5C Figs). Overall, properties of the ATAC-seq peaks observed for our four target tissues are comparable to those observed in the Drosophila literature [56], giving us confidence that the peaks of this study are robustly inferred. Finally, there is some suggestion that more highly enriched peaks (e.g., those near TSSs) tend to be more likely to be shared among tissues. S6 Fig shows the degree of peak sharing among tissues as a function of the feature type that peak is located in.

Our next goal was to obtain a common set of genomic locations (or loci) at which statistical tests to evaluate variation in chromatin accessibility over genotypes and/or tissues could be carried out. To do this we merged peaks (*i.e.*, the single base position where coverage peaked) over all tissues and genotypes that were within 200-bp of one another, and whose MACS2-defined boundaries overlapped. In contrast, peaks that were separated by more than 200-bp were not merged even if their MACS2 boundaries overlapped. To illustrate the merging procedure Fig 2 (top panel) depicts a representative ~30kb region centered on the gene *hairy* (a gene contributing to embryonic segmentation and peripheral neurogenesis) showing peaks called separately for each of the four tissues, as well as the consensus set of peaks with adjacent peaks merged (the "all tissues" track; see methods). Red hashes show the location of each peak, and horizontal black bars depict the entire peak interval from MACS2. The lower panel zooms in on a smaller 10kb region with a more detailed depiction of the raw coverage data (the y-axis is fold enrichment). As with typical ATAC-seq datasets we often see a strong peak near the TSS that is consistently identified across tissues. In contrast, for non-TSS peaks, MACS2 boundaries may only sometimes overlap depending on the tissue. The lower panel illustrates how our heuristic merges peaks close to one another across tissues to define a single peak location (red hashes). The heuristic gives a single "all tissues" location for the peak associated with the TSS of *hairy*, despite the peak position varying slightly among tissues. Furthermore, consider the region downstream of the 3' UTR of *hairy*, the MACS2 boundaries (indicated by the black bars) for two peaks overlap for ovaries, but not for the two disc tissues, and the six peaks each have different locations. Despite the MACS2 boundaries overlapping in ovaries, the raw coverage clearly suggests two peaks. As those peaks are greater than 200bp apart, the heuristic calls two peaks and further merges the positions of those two peaks across tissues. An algorithm that merges peaks based on overlapping boundaries, especially when data is collected from multiple tissues, would merge these two peaks (since their boundaries overlap), despite evidence they are separate. Based on visual inspections of the fold-enrichment profiles for many other regions (not shown) we observe many such instances where merging peaks based on overlapping MACS2 boundaries, especially those observed only in a subset of tissues, seems misleading, whereas keeping the peaks separate appears correct.

## Normalizing coverage corrects for sample-to-sample variation and the presence of structural variants

Different samples yield different numbers of raw reads. Additionally, histograms of ATAC-seq fragment lengths show a characteristic periodicity representing nucleosome free DNA, mono-

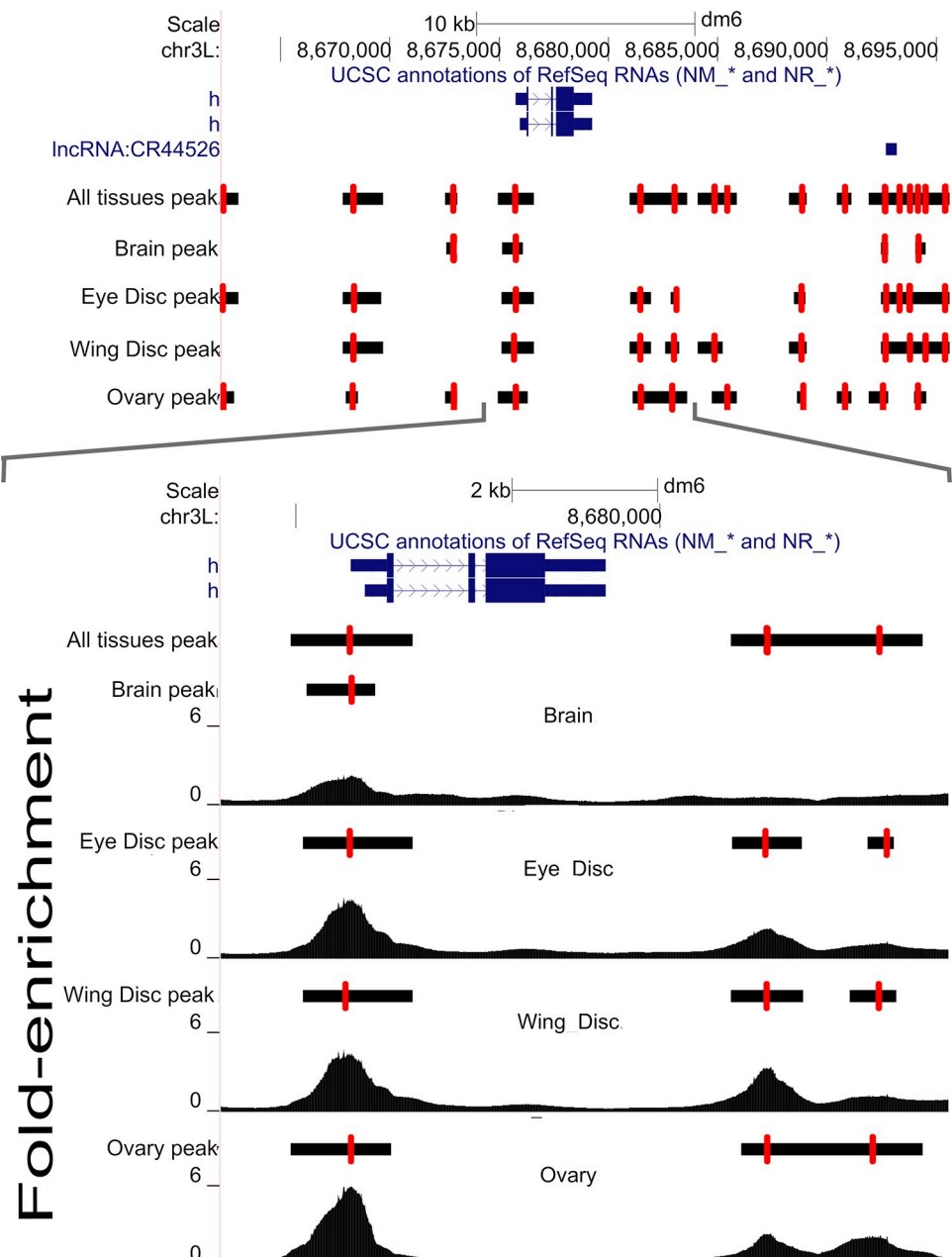

**Fig 2. An illustrative example of peak calling results near the gene hairy.** (Top panel) Peaks called separately by tissue as well as a consensus set of peaks calls (labeled "all tissues"). Single base peaks are indicated with red hash lines with black bars representing uncertainty. (Bottom panel) a zoomed region showing peaks and raw read coverage.

nucleosome bound DNA, di-nucleosome bound DNA, and so on (see Figs 2 of [22] and S7 for convenience). Fig 3, depicts the distribution of raw fragment lengths for two biological replicates of brain tissue ATAC-seq from the A4 strain in red (*i.e.*, independent tissue dissections and library preps). It is evident that replicate 2 has more nucleosome bound DNA than replicate 1. We hypothesize that such differences might arise from subtle differences in sample prep that result in different rates of disassociation of nucleosomes from DNA, and this sample-to-sample variation is likely challenging to experimentally control for. To allow comparisons

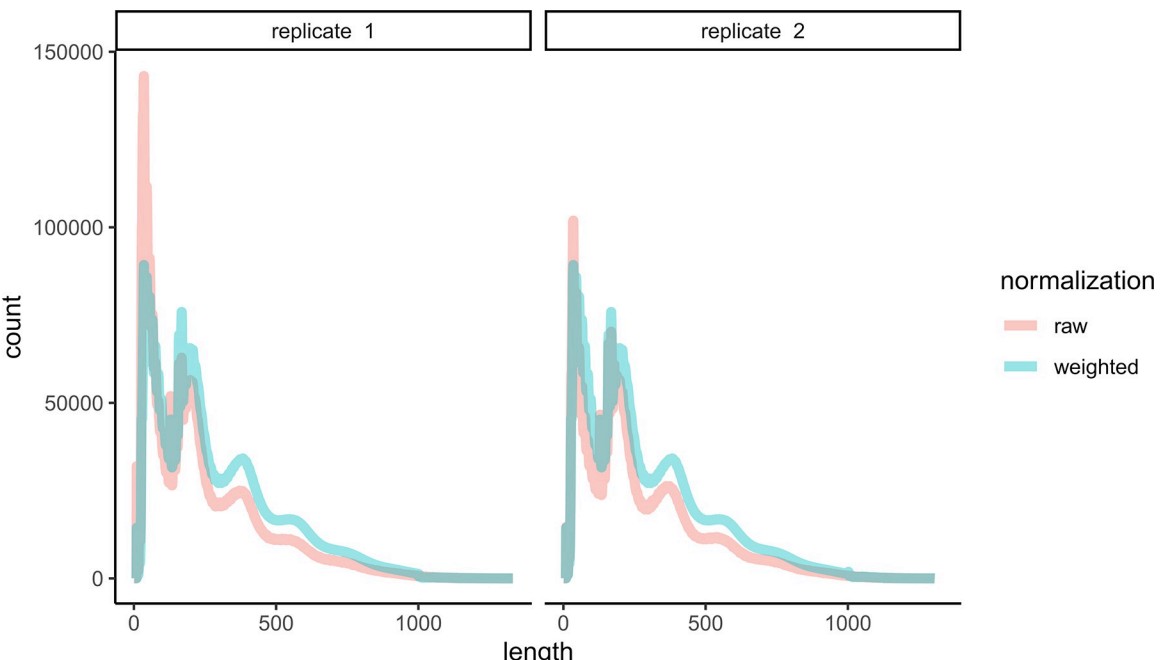

**Fig 3. Distribution fragment lengths before and after normalization.** Representative examples of the raw fragment size distribution for genotype A4 and brain tissue for two replicates in red. The same two samples are depicted in blue following normalization.

across tissues and genotypes we normalized each sample so that the genome-wide distribution of fragment sizes are identical (see methods) using an approach akin to the quantile normalization technique used extensively in the context of gene expression [57]. Our normalization results in a weight being assigned to each fragment and by working with those weights, as opposed to raw fragment counts, histograms have identical fragment size distributions across all samples (Fig 3, blue curves). This normalization allows for straightforward statistical testing between tissues and genotypes. S8 Fig depicts the distribution of fragment lengths across the 96 samples of this study prior to normalization and the removal of one sample due to low data quality.

A second concern often ignored in ATAC-seq analysis, that can make it difficult to compare samples across genotypes, is the presence of structural variants that could masquerade as polymorphisms in chromatin structure. ATAC-seq data obtained from different genotypes are generally aligned to a single reference genome, and a polymorphic structural variant near an ATAC-seq peak can result in unaligned reads, which will present as a local drop in coverage, and lead to the incorrect inference of more closed chromatin in that region of the genome. The eight genotypes examined in this study are highly isogenic and seven are associated with reference quality *de novo* assemblies, putting us in the unique position of being able to correct for polymorphic structural variants. We correct for SVs by excluding all fragments across *all* samples that span a structural variant present in *any* of the several assembled samples.

We illustrate the impact of correcting for SVs on wing disc ATAC-seq data for a 10kb region around the *rpr* gene (a gene important in programmed cell death) for two genotypes (B6 in brown, A4 in green), B6 harbors a ~17kb *mdg1* transposable element ~5kb upstream of the TSS of *rpr* (Fig 4A). In the uncorrected for SVs wing disc dataset, there is an apparent difference in chromatin configuration near the *mdg1* insertion. But after correcting for reads mis-mapped due to the *mdg1* insertion it appears that such an inference is incorrect and the lower coverage in the B6 genotype is largely due to mis-mapped reads associated with the

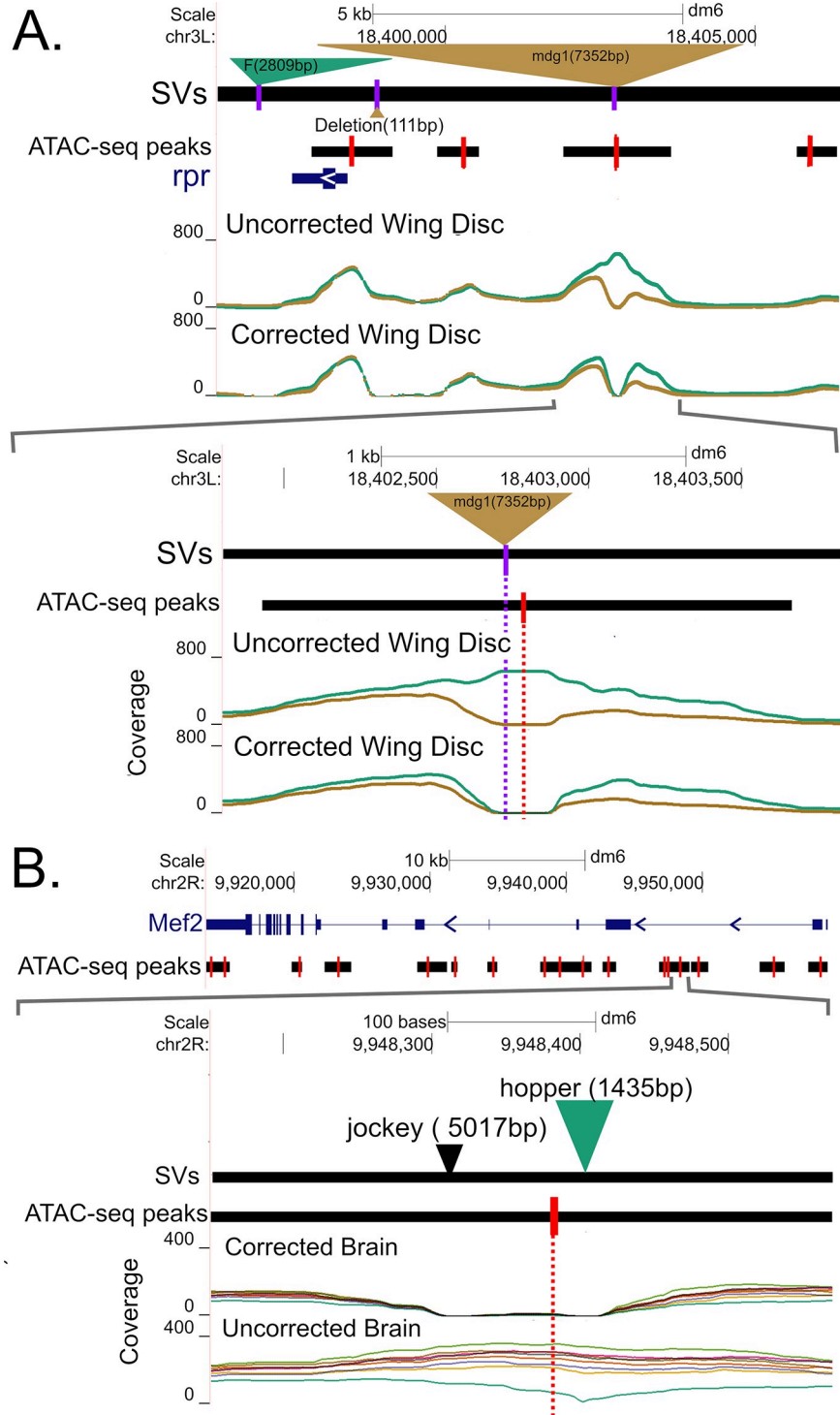

**Fig 4. Examples illustrating the effects of SV correction on coverage.** (A) After correcting for a large insertion of a mdg1 transposable elements upstream of rpr in strain B6 (brown) the apparent difference in coverage between strains B6 and A4 (green) is largely eliminated. (B) Correcting for the effect of a hopper TE in an intron of Mef2 in the A4 genotype largely eliminates an apparent difference in chromatin configuration.

*mdg1* TE. Although, even after correcting for the *mdg1* insertion, there does appear that there is a subtle difference in chromatin structure to the right of its location. Interestingly, this region contains two other SVs (a 2.8 kb *F* insertion in A4 and a 111bp deletion in B6) that do not impact chromatin structure inferences.

Fig 4B depicts a second example of an ATAC-seq peak in the first intron of the *Mef2* gene, whose product is crucial in myogenesis [58]. The top panel shows a 45kb region centered on the *Mef2* gene, while the bottom panel zooms in on a 550bp region entirely contained within the first intron showing coverage for brain samples with and without SV correction. In the SV-uncorrected data, this peak significantly varies by genotype with a $-\log_{10}$(FDR p-value) of 3.7. Seven of the genotypes exhibit a relatively open chromatin configuration, while A4 (in dark green) exhibits lower coverage in a region that contains a TE insertion. In a typical experiment, where the existence of the TE insertion would be unknown, and "hidden" from short read callers, the effect on read mapping of the TE would not have been corrected for, and we would have incorrectly inferred a genetic difference in chromatin accessibility. After SV correction, the ATAC-seq peak is not identified as being polymorphic. It is important to note that our correction acts by masking regions close to SVs in non-SV containing samples, so our proposed solution is far from perfect. But uncharacterized structural variants in non-reference genotypes can clearly cloud the interpretation of ATAC-seq datasets (as we show below).

Although both panels of Fig 4 illustrate transposable elements, other structural variants can impact read mapping. S9 Fig depicts a polymorphic 1.9kb deletion relative to the reference in strain A4 in the first intron of the *Abl* gene. The deletion knocks out two ATACseq peaks, but if only mapping short reads to the reference strain, A4 would appear to have a closed chromatin configuration.

## ANOVA identifies polymorphic chromatin structures

For every merged-peak in the euchromatic genome we carried out an ANOVA to determine if chromatin accessibility varies across Tissues, Genotype, or their interaction (T:G). We carried out this analysis for data either corrected or uncorrected for polymorphic structural variants for the seven genotypes with reference genomes. As the statistical analysis involved roughly sixty-eight thousand peaks and three *p*-values for each peak (Tissue, Genotype, and their interaction) we convert *p*-values to a false discovery rate and consider a test significant if the FDR is less than 0.5%. Table 1 gives the number of significant chromatin profile differences by factor, and Fig 5A shows tissue overlap using a Venn diagram. A robust observation is that for the SV-corrected data, close to 100% of all peaks display differences in chromatin features among the four tissues we examine. Of the peaks showing differences between tissues ~84% are not significant for a genotype or tissue by genotype interaction (Fig 5A). Thus, chromatin features are far more likely to vary among tissues than genotypes. Although differences between genotypes are far less frequent than differences between tissues, we still identify roughly 1000 such peaks (Table 1). Interestingly we identify roughly four times as many tissue by genotype interactions than simple genotype specific peaks. Finally, we created Manhattan plots for all

**Table 1. Number of peaks showing significant variation at an FDR of 0.5%.**

| Statistical Test | SV-corrected | SV-uncorrected |
|:---:|:---:|:---:|
| Genotype | 1050 | 2456 |
| Tissue | 30383 | 34361 |
| Genotype:Tissue | 4508 | 5792 |
| Total | 31769 | 36059 |

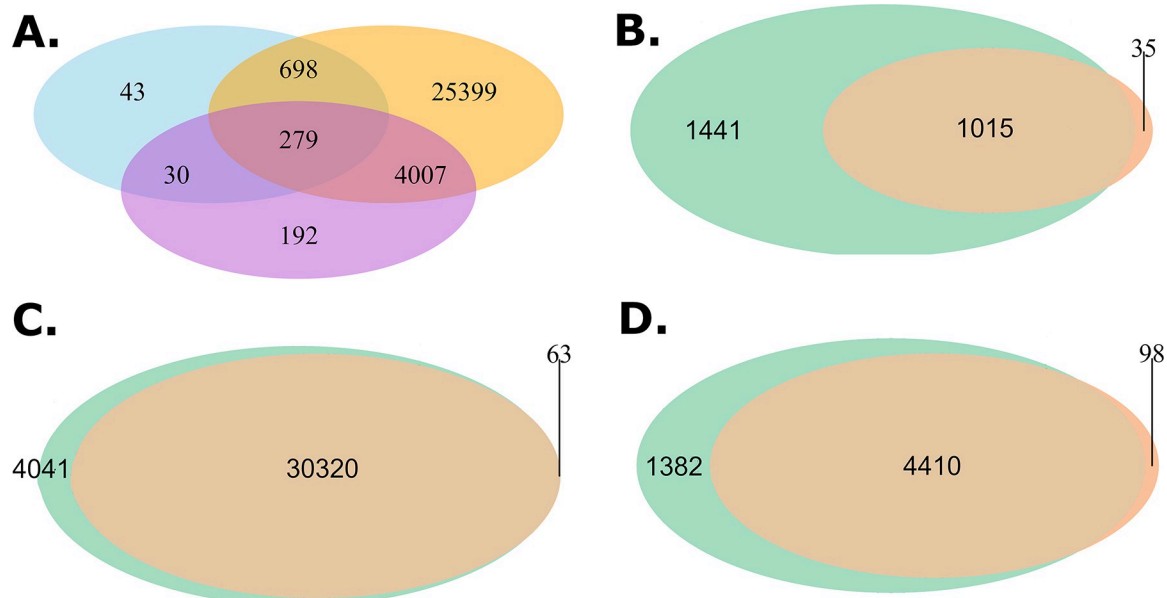

**Fig 5. Venn diagrams showing overlapping peaks by ANOVA categories and SV correction status.** (A) Venn diagram showing overlap of regions significant (FDR < 0.5%) for Genotype (blue), Tissue (orange), or G:T (orchid) for the SV-corrected data. (B-D) Venn diagrams showing the number of peaks significant G,T, or a G:T interaction, respectively. Green are tests carried out without correction for known SVs, and brown after SV-correction.

ANOVA tests, and observe that SV-corrected "hits" are largely uniformly distributed throughout the euchromatic genome with no evidence for "hotspots" (S10 Fig), although perhaps there is a tendency for an increased rate of significant genotype hits nearer centromeric regions (despite aggressive filtering for euchromatin only regions).

Since we carried out ANOVA on both SV corrected and uncorrected data, we can assess the impact of failing to correct for SVs on inference. There is considerable overlap in those peaks showing tissue-only effects between the uncorrected and SV-corrected datasets (Fig 5C). In contrast, we observe many fewer ATAC-seq peaks following SV correction in the genotype and tissue-by-genotype peak sets (Fig 5B and 5D): Of the peaks identified in the uncorrected analysis, 55% for the genotype-only set, and 21% for the tissue-by-genotype set are eliminated by correcting for SVs. We more carefully examined the peaks eliminated by SV correction ($n = 1441$ genotype-only, $n = 4041$ tissue-only, $n = 1382$ interaction) to determine what might be driving their disappearance (Table 2). The vast majority of these peaks—89%, 99%, and 90% for genotype, tissue, or the interaction, respectively—are either completely contained within an SV or are within 800bp of an SV boundary (Table 2). The location of these peaks suggests that they are purely the result of incorrect mapping of short sequencing reads from a non-reference genotype to a common reference genome. The remaining genotype- and

**Table 2. Number of peaks that are only significant in SV uncorrected data as a function of statistical test and distance from nearest SV.**

| Statistical Test | Number of Peaks | | | |
|---|---|---|---|---|
| | within | ±800bp | > ±800 bp | Total |
| Genotype | 801 | 481 | 159 | 1441 |
| Tissue | 2282 | 1736 | 23 | 4041 |
| Genotype:Tissue | 639 | 599 | 144 | 1382 |

interaction-only peaks that disappear following SV correction, but that are greater than 800bp from an SV, appear to be excluded by just failing to survive thresholds. Either they are eliminated by having their average coverage drop just below our threshold of 50 following SV-correction, or by just failing to reach our 0.5% FDR threshold in the SV corrected dataset (S11 Fig). Failing to correct for SVs during ATAC-seq peak calling—as is the norm when *de novo* genome assemblies are not available for the target strains—will generate large numbers of false positive peaks that do not, in truth, impact chromatin accessibility.

S12 Fig depicts false positive differences between genotype, tissues, or a genotype by tissue interaction as a function of the SV-type corrected for, and if the ATACseq peak is inside the SV or instead within 800-bp of an SV. In the case of indels, for example, an ATACseq peak could be contained within a deletion present only in one of the non-reference strains. In contrast, for a peak to be within a TE, that TE would need to be present in the reference strain and absent in the other strains examined, due to the way mapping to a reference genome works. Chakraborty *et al.* [43] observed 7347 TE insertions, 1178 duplication CNVs, 4347 indels, and 62 inversions in the euchromatin genomes of DSPR strains based on de novo sequencing. As expected, TEs dominate false positives due to SVs within 800bp of an ATACseq peak, whereas INDELs dominate the landscape for peaks contained within an SV. In general, the likelihood of a false positive is a complex function of the type of event, its population frequency, and how that event presents to short read mappers relative to the reference genome.

## Examples of polymorphic chromatin structures

Fig 6A depicts SV-corrected coverage brain and ovary samples centered between the TTS of the *Npc2f* gene, whose human ortholog (*NPC2* gene) is implicated in Niemann-Pick disease and Niemann-Pick disease type C2 due to its involvement in regulating sterol transport [59], and the TSS of *Kal1* gene, whose human ortholog (*Anosmin-1* gene) is responsible for the X-linked Kallmann's syndrome [60]. We observe a genotype polymorphism in chromatin state with the B2 genotype (light green) exhibiting a more closed chromatin state compared to the other genotypes for ovary tissue (-$\log_{10}$(FDR p-value) = 3.6). This ATAC-seq peak is further polymorphic by tissue with brain tissue exhibiting a generally closed chromatin state. We speculate that the B2 genotype has lower expression of *Kal1* in ovaries, with the chromatin

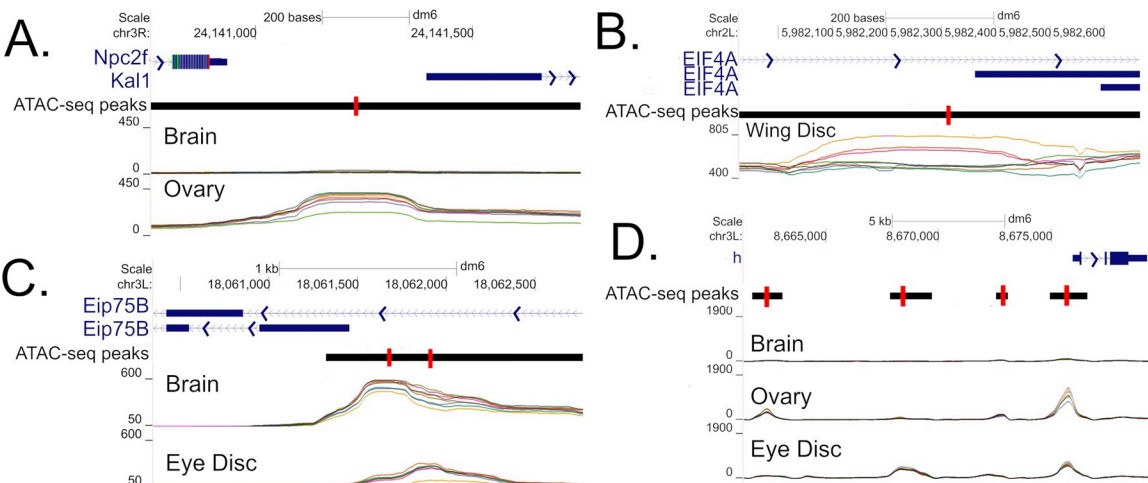

**Fig 6. Illustrative examples of polymorphic chromatin configuration.** The images depict regions upstream of the TSS of Kal1 (A), upstream of the TSS of a Eip75B isoform (B), upstream of the TSS of a Eip75B isoform (C), and a large region known to harbor cis-regulatory element upstream of hairy (D). SV-corrected coverage is given for a subset of interesting tissue.

structure impacting its TSS. Fig 6B depicts a polymorphic ATAC-seq peak located within the intron or near the TSS of the *eIF4A* gene (depending on isoform), with *eIF4A* acting as RNA-dependent ATPase and ATP-dependent RNA helicase that facilitates attachment of the 40S ribosomal subunit [61]. Coverages are higher for A5, A7, and B3, than the other genotypes in wing disc tissue (-$\log_{10}$(FDR p-value) = 3.5) with other tissues (not shown) showing similar trends in coverage. The location of the peak suggests a role in mediating isoform usage between genotypes via an alternative TSS, with the peak heights suggesting an allelic series. Fig 6C is an example of two adjacent peaks exhibiting a genotype:tissue interaction (-$\log_{10}$(FDR p-value) = 4.5 and 4.4 respectively) located in intron 1 of the *Eip75B* gene isoform F, and near TSS of *Eip75B* gene isoform E. This gene has been shown to regulate the complex traits of feeding behavior, fat deposition, and developmental timing [62,63,64]. As with the example of *EIF4A* we speculate that this polymorphism impacts isoform usage via alternative TSSs. Fig 6D depicts four peaks polymorphic by tissue, or by genotype:tissue interaction, for an interesting 14kb region directly upstream of TSS of *hairy*. *hairy* is well studied in the context of developmental biology [65–67] and the genetics of complex traits [68–70], with several *cis*-regulatory enhancers in this region playing a role in regulating the seven stripes formed in the blastoderm stage [71,72]. The four ATAC-seq peaks exhibit chromatin configurations that vary among tissues (-$\log_{10}$(FDR p-value) = 14.7, 17.9,10.9, and 14.9 left to right). Finally, the peak at chr3L:8672906 is polymorphic for a genotype:tissue interaction (-$\log_{10}$(FDR p-value) = 2.3).

## Candidate causative SNP identification

For each of the SV-corrected peaks significant for a genotype or genotype:tissue interaction we estimate the proportion of variation in peak height explained by each SNP (or marker) within 250bp of the peak (Fig 7). We speculate that such SNPs are strong candidates for *cis*-regulatory

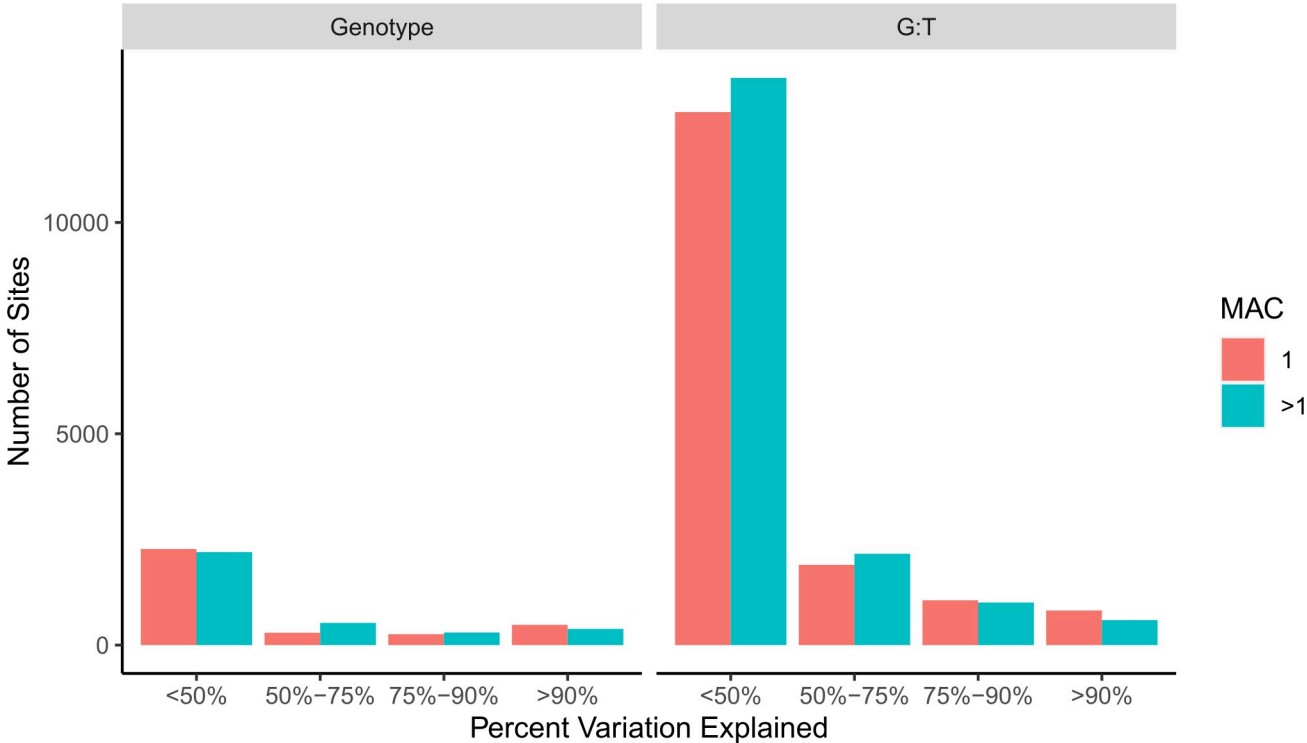

**Fig 7. ATAC-seq peak coverage variation explained by nearby polymorphisms.** Peaks significant for genotype or genotype by tissue are on the left and right respectively. The number of sites are grouped by Minor Allele Count (MAC).

Table 3. **Number of SNPs within 250 bp and explaining $\geq$ = 80% of the variation in coverage for peaks significantly varying by Genotype or G:T.**

| | Peaks that vary by: | |
| --- | --- | --- |
| Tissue | Genotype | Genotype:Tissue |
| Genotype | 1253 | NA |
| G:T | NA | 2735 |
| Brain | 1620 | 6485 |
| Ovary | 1299 | 5441 |
| Eye Disc | 1401 | 6780 |
| Wing Disc | 1465 | 7385 |
| Total Tests | 6707 | 33570 |

factors that control chromatin configuration. Two caveats are that we are only examining seven reference genomes so our models may be over-fitted, and truly implicating events as causative would require gene replacement experiments. We identify and test 6707 and 33570 SNPs located within 250bp of genotype or genotype: tissue interaction specific ATAC-seq peaks respectively. Out of those, there are 1253 (18.7%), and 2735 (8.1%) SNPs that explain greater than 80% of the variation in peak heights due to Genotype or G:T respectively, an average of 6 nearby SNPs per significant peak (Table 3). We further annotate all SNPs that explain 100% of variance for functional impact. Out of 687 SNPs that explain 100% of variance in peak height (by genotype or for a genotype: tissue interaction), there are a total of 22 SNPs annotated as having a high functional impact (i.e., missense, premature start codon, or splice variant), which is odd given that there is no reason to think a mutation of high functional impact on a transcribed protein is likely to impact a nearby chromatin configuration. The potential functional impact of the remaining 665 SNPs is more difficult to discern (S13 Fig).

## Examples of potentially causative SNPs

Fig 8A depicts an ATAC-seq peak downstream of TTS of *Bre1* isoform A and exon 4 of isoform B, a gene involved in regulation of *Notch* signaling [73–75]. Genotypes B2 and B3 are more closed in eye disc and brain compared to all other genotypes (Fig 8A). There is a potential causal SNP almost centered on the peak explaining 100% and 57% of the variation in eye disc and brain respectively. Fig 8B depicts a polymorphic peak for brain and ovary in which the A6 (purple) genotype appears more open than the others (-log$_{10}$(FDR p-values) = 6.1 and 10.2 for genotype and G:T respectively). The peak is located in a intron of *Ptpmeg* (involved in the maintenance of axon projection [76] and inhibition of EGFR/Ras/mitogen-activated protein kinase signaling pathway during wing morphogenesis [77]), as well as ~400bp downstream of TTS of *mthl9* (whose gene subfamily plays important role in *Drosophila* development, stress response, and regulation of life span [78]). Two nearby SNPs each explain 100% of variation in genotypes, and both are private to the A6. Fig 8C depicts a peak polymorphic by genotype that appears largely brain specific with A7 (pink) being more closed relative to other genotypes, and B2 (light-green) perhaps more open slightly downstream but not associated with a called peak. A nearby SNP private to A7 in the 5'-UTR (and 51bp downstream of a TSS) of a *Nna1* isoform explains 99% variance in genotype in brain. Fig 8D depicts potentially causal SNPs exhibiting a genotype:tissue interaction located upstream of two TSSs for the gene *stv* (involved in the chaperone pathway essential for muscle maintenance [79]). For both peaks and tissues the A4 (green) genotype exhibits a more closed configuration especially in wing disc and to a lesser extent brain. A SNP private to A4 explains 81%, 95%, 55%, and 97% of the variance in coverage for brain and wing disc at left and right peaks respectively.

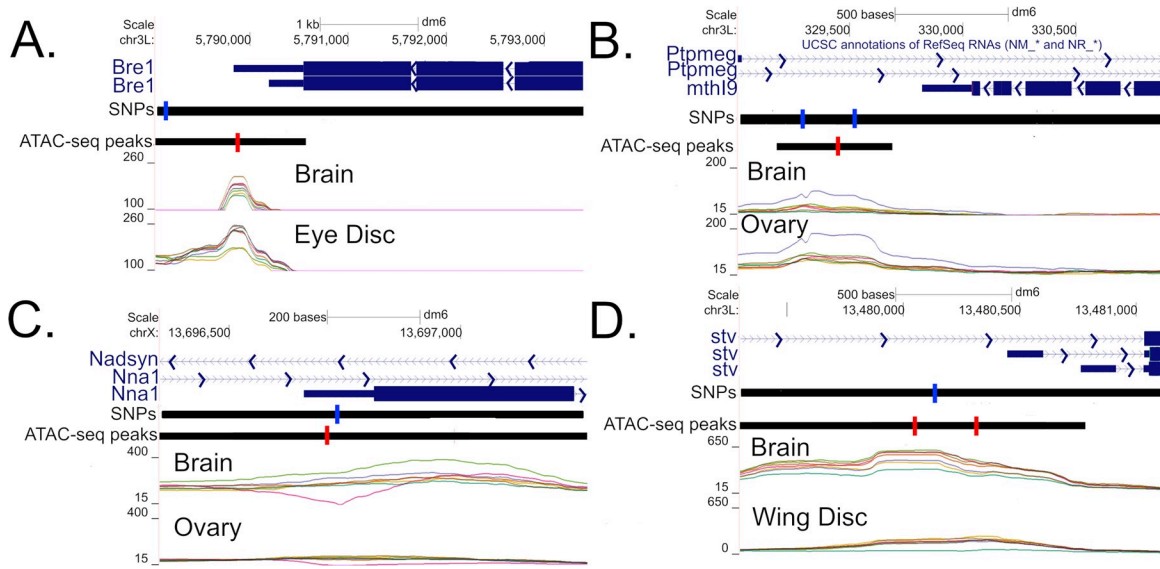

**Fig 8. Illustrative examples of putatively causal SNPs.** Regions are depicted downstream of the TTS of Bre1 (A), downstream of the TTS of mthl9 (B), the 5'UTR of a Nna1 isoform (C), and upstream of the TSSs of two stv isoforms (D). Only shows SNPs explaining > 80% of variation in Genotype of a G:T interaction (blue) are depicted.

## Discussion

Previous ATAC-seq/DNase1-HS-seq experiments in *Drosophila* have focused almost exclusively on embryos, whole adult bodies, or cell lines, and only rarely have compared multiple genotypes. We carried out a replicated ATAC-seq experiment on two adult tissues (female brains and ovaries) and two imaginal disc tissues (wing and eye-antennal imaginal discs) from which adult tissues are ultimately derived. It is widely believed that the sites that contribute to complex trait variation are likely to be regulatory in nature, thus chromatin features expressed in adult tissues are strong candidates to harbor such causative sites. Thus, we expect the data collected as part of this experiment will be of utility to the *Drosophila* complex trait community, who tend to study traits that manifest in adult or larval flies (c.f. Table 3 of Mackay and Huang 2018) [45], and see utility in our distributing coverage as a function of genotype and tissue as a series of Santa Cruz Genome Browser tracks (http://goo.gl/LLpoNH). We characterize eight highly isogenic strains of *Drosophila* that are a subset of the strains used to found the *Drosophila* Synthetic Population Resource [48], with seven of those strains having reference quality genome assemblies levels [43]. We largely employ a standard ATAC-seq peak calling pipeline, apart from our strategy for merging peaks within and between tissues, to obtain a union dataset consisting of 44099 open chromatin peaks.

Our analyses identified approximately thirty thousand peaks that differed in coverage between tissues, highlighting the future need for tissue specific chromatin maps. We further identified on the order of one thousand chromatin peaks that differ among genotypes and five thousand that vary among genotypes in a tissue specific manner. Chromatin peaks that differ among genotypes associated with candidate genes identified via QTL mapping in DSPR [48] or GWAS using DGRP [80] are strong candidates for contributing to differences in gene expression levels. Surprisingly, peaks displaying genotype by tissue interaction are more frequent than the genotype specific peaks. Such peaks represent candidates for modulating gene expression in a genotype dependent manner in a small subset of tissues that gene impacts. It is reasonable to speculate that ATACseq peaks displaying tissue by genotype interactions

underlie QTL that appear to be tissue or complex trait specific and do not show a great deal of pleiotropy.

We carried out statistical testing to identify chromatin states that vary among tissues, seven of the eight wild-type genotypes, and/or exhibit a tissue by genotype interaction (*i.e.*, differences among genotypes varying in the tissue dependent manner). To facilitate statistical testing, we carried out two important data normalization steps unique to this study. We first developed a per sample normalization procedure that creates per fragment weights that control for differences between samples in the total number of reads, and the percentage of read pairs that are nucleosome-free, mononucleosomic, binucleosomic, etc. The degree to which normalization impacts inference depends on how similar the fragments distributions are between samples. Some tissues seem easily amenable to ATACseq preps, especially cell lines, in which case perhaps no correction is necessary. On the other hand, more difficult tissues, will result in larger differences in fragment size distributions, and the correction is more likely to be beneficial. Any normalization method is likely to be most useful when comparing genotypes within tissue, where subtle differences in ATACseq peak heights could be biologically meaningful. There is some risk that differences fragment size distributions between tissues could be biological in origin, a problem shared among all between tissue normalization methods. These caveats acknowledged, the observation of fragment size distribution differences between biological replicates suggests that normalization may be beneficial.

A second important normalization step attempted to control for false positive inferences due to hidden structural variants. By virtue of seven of the eight isogenic strains being associated with reference quality *de novo* assemblies [43], we control for the potential artifact of polymorphic structural variants creating read-alignment differences that in turn could masquerade as differences in chromatin configuration. We accomplish this by masking regions in all strains harboring a nearby SV present in any strain. We carried out statistical testing on datasets either ignoring or following correction for polymorphic SVs and estimate the potential to identify false positives in data sets where SVs are hidden. Failure to account for SVs does not strongly impact the inference of differences in coverage between tissues, but it can have a huge impact in terms of detecting difference in chromatin accessibility between genotypes or those showing a genotype by tissue interaction, where we estimate potential false positive rates of 48% and 19% respectively. Our method of correcting for SVs is conservative and consists of masking regions association with polymorphic SVs.

A shortcoming of our masking SVs is that we cannot perform in depth analyses of possible biological effects of the SVs themselves (unless they exert those effects over distances longer than ~800bp). A potential solution would be to align reads to a genome private to each strain, followed by lifting those alignments over to a universal coordinate system to compare genotypes. Although this approach works well for SNP-based variation in well-behaved genomic regions, we find that lift-overs tend to break down when structural variants distinguish strains [81], and this is especially problematic for events like duplications where there is not even a 1:1 mapping between genomes. While our method of masking SV is not perfect, it is simple to implement and can remove upward of 50% of false positive peaks.

For ATAC-seq peaks that vary significantly in coverage among genotypes or that show a tissue by genotype interaction we attempted to identify nearby SNPs (or markers) that may control that variation. It is both reasonable to suggest, and supported by experiments (c.f., [24,82–85]), that alleles that control chromatin accessibility peaks are likely to be in *cis* and physically close to the peak. We identify several thousand such SNPs that explain more than 80% of the variation due to genotype or a genotype by tissue interaction for coverage, a collection likely enriched for causative polymorphisms, despite our over-fitting of the data. It would be of value to extend this approach to a much larger collection of genotypes, although such work may

necessitate focusing on a single tissue and require more *de novo* genome sequences to control for hidden structural variants. As Crispr/Cas9/allele swapping methods continue to come of age in *Drosophila* [86–90] medium-throughput functional assays capable of confirming specific allele chromatin peaks interactions could alternatively be used to characterize alleles regulating nearby chromatin states and gene expression levels.

## Materials and methods

### Strains

We employed 8 of the 15 strains that serve as founders of the *Drosophila* Synthetic Population Resource (DSPR), a multiparental, advanced generation QTL mapping population consisting of hundreds of recombinant inbred lines [48]. These highly-inbred strains—A4, A5, A6, A7, B2, B3, B6, and B7 (S1 Table)—are a worldwide sample of genotypes (S1 Fig), and seven of the eight (excluding B7) have reference quality assemblies such that virtually all SVs are known [43].

### Tissue dissection and ATAC-seq library preparation

The 8 inbred strains were raised and maintained in regular narrow fly vials on a standard corn-meal-yeast-molasses media in an incubator set to 25˚C, 50% relative humidity, and a 12 hour Light: 12 hour Dark cycle. We isolated nuclei from four different tissues for our 8 target strains. (1) Adult brains (central brain + optic lobes) were dissected and pooled from ten 1–4 day old females per replicate. (2) Ovaries were dissected and pooled from five 1–5 day old females per replicate. (3) Wing imaginal discs were dissected and pooled from 3–7 male wandering third instar larvae per replicate. (4) Eye-antennal imaginal discs were dissected and pooled from 4–7 male wandering third instar larvae per replicate. For each strain/tissue combination we generated 3 replicates. All dissections were carried out 1–9 hours after lights on, and following dissection all samples were immediately subjected to nuclei isolation.

Our full protocol for ATAC-seq library construction is provided in S1 Text, but in brief: Animals were dissected in nuclei lysis buffer under a standard stereoscope, and dissected tissue for a given replicate pooled into 200-µl of nuclei lysis buffer on ice. Each sample was then subjected to manual grinding, passed through 30-µM filter cloth, spun down, and the supernatant removed. Subsequently, 25-µl of tagmentation reaction mix was added to the pellet, and incubated at 37˚C for 30-min before freezing at −20˚C. After thawing, the sample was cleaned using a MinElute PCR purification column (Qiagen, 28004), and an aliquot was subjected to PCR to add on custom, Illumina-compatible indexing oligos. Finally, samples were cleaned using a standard bead-based approach, quantified using a Qubit dsDNA BR kit (Thermo-Fisher, Q32850), and examined via a TapeStation 2200 using genomic DNA ScreenTapes (Agilent Technologies, 5067–5365 / 5067–5366).

All 96 libraries (8 strains × 4 tissues × 3 replicates) were pooled at equal amounts—along with a series of other libraries that are not part of the project—and run over 16 lanes of an Illumina HiSeq4000 sequencer at the UCI Genomics High-Throughput Facility collecting PE50 reads.

### Read processing

Adapters were trimmed from the raw reads using Trimgalore-0.4.5 [91,92], and trimmed reads were aligned to the dm6 *D. melanogaster* reference genome [93] using bwa 0.7.8 [94]. Unmapped reads, and reads with unmapped mates were removed with samtools 1.3 (option -F 524 -f 2) [95], and all non-primary reads and improperly aligned reads were also removed with

samtools 1.3 (option fixmate -r and option -F 1084 -f 2). Following this, duplicate reads are removed using picard 2.18.27 [95,96] via MarkDuplicates and REMOVE_DUPLICATES = TRUE. Only reads aligning to the five major chromosome arms —X, 2R, 2L, 3R, and 3L - were retained for analysis. BAM files were corrected to reflect the actual insertion points of the Tn5 transposase acting as a dimer by having plus strand reads shifted +4bp and minus strand reads shifted −5bp as suggested in [22]. We refer to these as "corrected BAM files". Paired end BED files reflecting mapped fragments were generated using bedtools 2.25.0 [97]. The same process was carried out for all 96 samples (8 genotypes, 4 tissues, and 3 replicates).

## ATAC-seq peak calling

Corrected BAM files from all 96 samples were merged by tissue across replicates and genotypes, and MACS2 [49] was used to call peaks separately on the ovary, brain, wing disc, and eye disc. MACS2 options were -f, -p 0.01 to set cut-off p-value for peaks to be considered significant, -B—SPMR,—no-model to skip any read shifting as we were using corrected BAM files, and -g was set to 142573017, the summed length of the major chromosome arms in the dm6 genome release. The peak calling resulted in four ENCODE "tissue NarrowPeak files", one for each tissue.

## Merging of peaks across tissues

Tissue NarrowPeak files were concatenated, sorted by chromosome and peak summit, then a custom python script grouped and averaged peak summit locations that were within 200 bp of one another, but greater than 200 bp from the nearest adjacent peak summit. Each averaged peak summit is associated with a minimum left interval boundary and maximum right interval boundary obtained from all the summit peaks contributing to an average peak. The resulting file was converted to ENCODE NarrowPeak format for viewing using the UCSC genome browser with "peak" as peak name,"1000" as peak score, "." as peak strand, "10" as peak enrichment, and "-1" as q-value and p-value to accommodate the ENCODE NarrowPeak format, referred to as the "all tissue" track/peak file. Only the chromosome and the mean peak summit columns are used in downstream statistical analysis steps.

## Euchromatin peak filter and peak annotation

We choose to focus solely on euchromatic regions of the genome since heterochromatic regions are gene poor, poorly annotated, and enriched for structural variants and transposable elements. The euchromatin region boundaries we employ are given in S2 Table and come from [98]. All peaks in the all tissue peak file, and the four tissue NarrowPeak files used in downstream analyses, include only euchromatin located peaks.

We used HOMER v 4.11.1 [99] and the tissue NarrowPeak files separately for each of the four tissues to annotate each peak summit as belonging to one of eight exclusive groups based on their location relative to features annotated in the dm6 reference genome: (1) transcription start site (TSS: −1000 to +100bp from the transcription start site), (2) transcription termination site (TTS: −100 to +1000bp from transcription termination site), (3) coding exons, (4) 5' UTR exon, (5) 3' UTR exon, (6) intronic, (7) intergenic, and (8) non-coding (referring to non-protein-coding, but nonetheless transcribed DNA). In the case of a peak belonging to more than one feature type it is assigned to a single feature type with priority according to the numeric order of the features in the previous sentence (*i.e.*, TSS has priority over 5' UTR, etc). Since we focus in this work on peak summits, whereas HOMER annotates a peak as being at the midpoints of an interval, we edited the interval associated with each peak to be the peak summit

+/− 1bp. The percentages of peaks falling into each feature type by tissue are given in the S3 Table. These percentages are compared to the annotation types associated with one million randomly assigned peak locations. Comparing these percentages between tissues and/or to other studies is a measure of quality control.

## Quality control of ATAC-seq peaks

We carried out manual quality control steps on the dataset. First, we generated Venn diagrams for each tissue to compare the number of peaks using two different cut-offs for significance in the MAC2 peak calling. We compared cut-offs of -log(p-value) > = 2 and > = 3 (MACS2 p-value cutoff suggestions) using R package VennDiagram version 1.6.20. The number of overlapping peaks were calculated via the mergePeak function in HOMER with option -venn on the tissue bed files. We forced the maximum distance between peak centers to be < = 100bp for two peaks to be considered "overlapping". We observed the degree of overlap to be qualitatively similar for -log(p-value) cutoffs of either 2 or 3, as a result we employ a cutoff of 2 for peaks called by MACS2.

We further created several plots using the peak fold-enrichment profiles obtained from MACS2. Peak fold-enrichments are a measure of read counts at peaks relative to the local random Poisson distribution of reads [49]. ATAC-seq peaks are typically highly enriched in transcription related genomic regions, such as TSS, TTS, 5' UTR, or exons [52,100]. We similarly examined fold-enrichment as a function of annotated region type to confirm our data is consistent with previous work. We similarly examined fold-enrichment profiles as a function of distance from the nearest TSS to ensure our peaks were consistent with prior work. We also looked at the distribution of fragments lengths for each library to make sure libraries were not dominated by naked DNA and exhibited peaks associated with nucleosome bound DNA (c.f. S7 and S8 Figs). Lastly, we generated Manhattan plots of peak locations to ensure that they are not spatially clustered at a gross genomic scale.

## Normalization for differences between tissues and genotypes

For the $j^{th}$ sample (*i.e.*, replicate/tissue/genotype combination) we have a "fragment file" generated from the corrected BAM file that is a 3-column BED file with the chromosome, corrected start and corrected stop base of each fragment defined by a set of paired reads. In order to carry out statistical tests at peaks using our replicated ATAC-seq data we normalized the 95 different fragment files associated with each tissue/genotype/replicate combination (as one sample failed a visual quality control check). Our normalization procedure is based on the observation that both the number of reads and the distribution of fragment lengths varies between samples (see Fig 3A and 3B). The former just reflects variation in the number of reads obtained per library, and we believe the latter is due to subtle differences in sample preparation that inadvertently selects for differing fractions of nucleosome free DNA. Our normalization consists of adding a 4th column to the fragment file that can be thought of as a "weight" used in all downstream analyses, where that weight normalizes the fragment files across the J samples. The weight is inspired by the "quantile normalization" method used in the field of gene expression [57] and is simply: $w_{ij} = N_{i\cdot}/N_{ij}$, where $N_{ij}$ is the number of fragments of length $i$ in the $j^{th}$ sample, and the "." is the average over samples. As can be seen from the unweighted and weighted histograms of Fig 3 these weights result in a distribution of fragment lengths that are identical between samples.

With weights in hand, we calculated the weighted Coverage for each sample at any given position in the genome, C, as the sum of the weights of all fragments covering that position. We finally averaged coverage over replicates (within tissues and genotypes) to generate UCSC

Genome browser tracks [101–104], although the biological replicates were retained for statistical testing.

## Accounting for structural variants

In a typical ATAC-seq experiment raw reads are aligned to a reference genome, fragment files are derived from those alignments, and the resulting fragment files are perhaps normalized. However, in the case of the seven strains (A4,A5,A6,A7,B2,B3,B6) of this study we have complete *de novo* reference quality genome assemblies [43]. The genomes of these strains are distinguished from the dm6 reference by thousands of SVs, such as mobile element insertions, smaller insertions or deletions of DNA sequences (large enough to generally not be identified by standard pipelines), tandem duplications, and inversions. These "hidden" structural variants can impact inferences regarding chromatin structure obtained from ATAC-seq data assembled to a standard reference. To illustrate this issue we simulated 50bp PE reads from a 30kb or 32.5 kb genomic region using samtools::wgsim, with the two sequences being identical aside from a 2.5kb insertion of DNA sequence derived from a transposable element. Simulated short reads are then obtained from each region with an average fragment length of 400bp (standard deviation of 100 bp), similar to the fragment length distribution of ATAC-seq reads. Reads were mapped back to the shorter region (akin to a "reference genome") and coverage is depicted in S14 Fig. The figure clearly shows the potential for mis-mapped reads associated with a polymorphic SV to create a large localized dip in sequence coverage (that could be interpreted as a closed chromatin structure), with the footprint of this phenomena likely restricted to +/- 800bp (i.e., 2 standard deviation in read length) around the SV.

In the work of this paper, by virtue of seven of the genotypes examined having reference quality *de novo* assemblies, we can control for the effect of unmapped reads due to structural variants by removing fragments *across all genotypes* that span an SV *in any given genotype*. This correction is done using bedtools intersect to remove all reads from all fragment files that span insertion or deletion variants. For duplication variants, we first calculated duplicated regions by adding and subtracting the total length of duplication from the insertion site. Then, all reads spanning duplicated regions are removed. We then calculate new weights as described above, and recalculate coverage.

## Statistical testing

We carry out two ANOVA statistical tests for seven strains with reference qualify assemblies (A4, A5, A6, A7, B2, B3, B6) at peaks to identify those that differ among genotypes, tissues, or their interaction for weighted log transformed Coverage ($lnC = ln(C+5)$) after peaks with a weighted average coverage < 50 were dropped as: *lnC ~ geno + tissue + geno:tissue*. Adding 5 to the number of counts makes the rare case of counts near zero less extreme relative to other strains. A False Discovery Rate (FDR) associated with each p-value was calculated using the p.adjust function in R [105,106]. Tests with FDR adjusted p-values < 0.005 (or $-\log_{10}$(FDR p-value) > 2.3) are considered significant. QQ plots and Manhattan plots were generated for the ANOVA results.

We carried out statistical tests on both SV corrected and uncorrected fragment files. Loci significant in the SV-uncorrected data but not significant in the SV corrected data potentially represent false positives. We define hits unique to the SV-uncorrected dataset as false positives and estimate the rate of such false positives in experiments that do not correct for hidden SVs. Results are also represented as Venn diagrams. We further examined each potential false positive to determine if the ATAC-seq peak was actually contained within a SV (e.g., a deletion relative to the reference), was within +/- 800bp of an SV boundary, or was >800bp from an SV

(for peaks >800bp from a SV both F&R reads are expected to map to the reference genome and thus such peaks are not expected to be impacted by the correction). We finally compared the p-values between SV-corrected and uncorrected data for peaks >800bp from an SV to determine if any remaining differences were due to simple sampling error in p-values near significant cut-offs.

## Causative SNP and SV identification by random effect model

For peaks significant for genotype or genotype:tissue we attempted to identify SNPs within 250bp of the peak that could potentially explain the significant result. We accomplished this via the following random effects model in R::lme4 (version 1.1–23): *lnC ~ (1|marker) + (1|marker:tis) + (1|tis) + (1|geno:marker) + (1|tis:geno:marker)*

We estimate the proportion of variance explained by a marker as $var_m/[var_m + var_{g:m}]$ or marker:tissue as $var_{m:t}/[var_{m:t} + var_{g:m:t}]$ respectively. Here marker refers to a state of a SNP, thus several genotypes could share the same marker state. Furthermore, since the strains of this study are isogenic, markers are either REF or ALT, and never heterozygous. In both cases a ratio close to 100% identifies a SNP that explains all the variation associated with a significant peak. We similarly estimate the proportion of variation explained for each tissue by dropping terms involve tissue. These SNPs are strong candidates for being causative, with the strong caveat that only 7 genotypes are examined in this study, so we are almost certainly over-fitting and confirmatory experiments are necessary. We examine the distributions of these marker tests and maintain a list of polymorphisms explaining 100% of the variation associated with peaks. We finally annotate SNPs explaining 100% variance using SnpEff [107] and HOMER. In addition, a list of SNPs/SVs which individually explain less than 80% variance of polymorphic peaks is also provided. These SNPs/SVs potentially explain only a fraction of the variation in peak height, with the remaining due to other cis-acting or trans-acting variants. Future confirmatory experiments are even more necessary to confirm the causal effect of these SNPs/SVs.

## Supporting information

**S1 Table. Details of the eight strains examined in this study.** All eight strains are P-element and Wolbachia free, were brother sister mated for up to 18 generations, and are highly isogenic [48]. Each strain, bar B7, is associated with a reference quality de novo genome assembly [43]. The Stock Number is the Bloomington ('b') or Tucson/San Diego ('t') Drosophila Stock Center code, although these strains are no longer available from these centers. The stock Full Name, if any, is also given.
(DOCX)

**S2 Table. Euchromatin boundaries employed in this work (dm6 coordinates).**
(DOCX)

**S3 Table. Raw euchromatin peak count by tissue for each feature type.** The Genome column is the percent of each feature type in the genome.
(DOCX)

**S4 Table. Mapping statistics.**
(DOCX)

**S1 Fig. World map showing the collection locations and color legend for all genotypes.** The color legend for genotypes is also kept constant throughout the paper. This map was created using mapchart.net, licensed under This work is licensed under a Creative Commons

Attribution-ShareAlike 4.0 International License (CC BY-SA 4.0).
(DOCX)

**S2 Fig. Workflow for ATAC-seq study.**
(DOCX)

**S3 Fig. Summary statistics for peaks called for ovary samples.** (A) Distribution of peak enrichment scores for the ovary samples. (B): Peak enrichment scores as a function of distance to the nearest transcription start site with a smoothing line for the ovary samples. Insert focuses on peaks within 10kb of the TSS and showing only the smoothing line. (C): Peak enrichment distribution as a function of genomic feature for the ovary samples.
(DOCX)

**S4 Fig. Summary statistics for peaks called for eye disc samples.** (A) Distribution of peak enrichment scores for the eye disc samples. (B): Peak enrichment scores as a function of distance to the nearest transcription start site with a smoothing line for the eye disc samples. Insert focuses on peaks within 10kb of the TSS and showing only the smoothing line. (C): Peak enrichment distribution as a function of genomic feature for the ovary samples for the eye disc samples.
(DOCX)

**S5 Fig. Summary statistics for peaks called for the wing disc samples.** (A) Distribution of peak enrichment scores for the wing disc samples. (B): Peak enrichment scores as a function of distance to the nearest transcription start site with a smoothing line for the wing disc samples. Insert focuses on peaks within 10kb of the TSS and showing only the smoothing line. (C): Peak enrichment distribution as a function of genomic feature for the wing disc samples.
(DOCX)

**S6 Fig. Peak sharing among tissues as a function of feature type.**
(DOCX)

**S7 Fig. Fragment length distribution and the nucleosome binding configuration depicted by the fragment length.**
(DOCX)

**S8 Fig. Fragment length distribution for all replicates, tissues, and genotypes.**
(DOCX)

**S9 Fig. A polymorphic deletion relative to the reference leads to the incorrect inference of close chromatin in strain??.**
(DOCX)

**S10 Fig. Manhattan plots showing that significant (FDR $< 0.005$) peaks do not show strong evidence for spatial clustering throughout the genome.** (A): Manhattan plot for significant peaks by genotype. (B): Manhattan plot for significant peaks by tissue. (C): Manhattan plot for significant peaks by genotype and tissue interaction.
(DOCX)

**S11 Fig. -log(FDR adjusted p-value) scatterplot comparison between SV-corrected data and SV-uncorrected data for false positive peaks that falls outside of SV affected regions.** Red dashed lines showing -log(p-value) = 2.3 for SV-uncorrected (horizontal) and SV-corrected (vertical). Sampling variation likely drives the observed differences, as hits tend to be just beyond the significance level in the uncorrected dataset.
(DOCX)

**S12 Fig. Number of false positive significant peaks within structural variants (left) and within 800bp of the structural variants (right) by statistical test carried out (Genotype, Tissue, or Genotype by Tissue).** Variant types are deletion relative to reference (DEL), insertion due to TE (TE-INS), non-TE insertions (Other-INS), inversion (INV), or copy number variant (CNV). The categories are non-exclusive as multiple SV events could be close to an ATAC-seq peak, especial since we integrate over all strains.
(DOCX)

**S13 Fig. SnpEff annotation for causative SNPs that explain 100% variation.**
(DOCX)

**S14 Fig. Example showing the effect of structural variant on inferred fragment coverage.** The green track depicts the reference 2kb sequence with the pink track depicting a non-reference 2.5kb transposon insertion (not drawn to scale) at the location of the red dash. The bottom plot depicts the coverage of reference (green) or non-reference sample aligned to the reference genome. The stars depict short read (forward and reverse reads) pairs and dashed lines the fragments created by read pairs. Chimeric fragments with one read in the TE insertion are mis-mapped, resulting in strong dips in read coverage at locations close to the TE insertion site.
(DOCX)

**S1 Text. ATAC-seq protocol.**
(DOCX)

## Author Contributions

**Conceptualization:** Khoi Huynh, Stuart J. Macdonald, Anthony D. Long.

**Data curation:** Khoi Huynh, Brittny R. Smith, Stuart J. Macdonald, Anthony D. Long.

**Formal analysis:** Khoi Huynh, Anthony D. Long.

**Funding acquisition:** Stuart J. Macdonald, Anthony D. Long.

**Investigation:** Brittny R. Smith, Stuart J. Macdonald, Anthony D. Long.

**Methodology:** Khoi Huynh, Brittny R. Smith, Stuart J. Macdonald, Anthony D. Long.

**Project administration:** Stuart J. Macdonald, Anthony D. Long.

**Software:** Khoi Huynh, Anthony D. Long.

**Supervision:** Stuart J. Macdonald, Anthony D. Long.

**Validation:** Khoi Huynh, Brittny R. Smith, Stuart J. Macdonald, Anthony D. Long.

**Visualization:** Khoi Huynh, Stuart J. Macdonald, Anthony D. Long.

**Writing – original draft:** Khoi Huynh, Anthony D. Long.

**Writing – review & editing:** Khoi Huynh, Brittny R. Smith, Stuart J. Macdonald, Anthony D. Long.

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
