## [Decision Letter · Decision Letter 0]

14 Dec 2022

Dear Dr Long,

Thank you very much for submitting your Research Article entitled 'Genetic Variation in Chromatin State Across Multiple Tissues in Drosophila melanogaster' to PLOS Genetics.

The manuscript was fully evaluated at the editorial level and by three independent peer reviewers. All three reviewers and the editors are enthusiastic about this manuscript. The dataset is timely, the manuscript is well written, and the analyses are insightful and informative. The reviewers all have concerns/suggestions about some of the analyses and interpretations that we hope will strengthen the paper when addressed.  Based on the reviews, we will not be able to accept this version of the manuscript, but we are willing to review a revised version. We cannot, of course, promise publication at that time.

If you decide to revise the manuscript for further consideration at PLOS Genetics, please aim to resubmit within the next 60 days, unless it will take extra time to address the concerns of the reviewers, in which case we would appreciate an expected resubmission date by email to plosgenetics@plos.org.

We are sorry that we cannot be more positive about your manuscript at this stage. Please do not hesitate to contact us if you have any concerns or questions.

Yours sincerely,

Kelly A. Dyer

Academic Editor

PLOS Genetics

Kirsten Bomblies

Section Editor

PLOS Genetics

Reviewer's Responses to Questions

**Comments to the Authors:**

Reviewer #1: Huynh et al conducted ATAC-seq across several tissues in 8 Drosophila strains with high quality genome assemblies to understand how chromatin accessibility at cis-regulatory regions differ across tissues and genotypes. Taking advantage of the high quality genome assemblies of these lines, they inferred that sizable numbers of ATAC-seq peaks (called by MACS) that vary between strains and tissues, are false positives due to structural variants absent in the reference genome. After accounting for these issues (typically due to mapping errors) they identified a high-confidence set of peaks that vary between tissues and genotypes.

Overall, I believe this study demonstrates a very important issue regarding the use of ATAC-seq, especially given how popular the method is as a means of assessing epigenetic and regulatory changes. This is particularly problematic when comparing between genotypes, but perhaps less so when comparing between tissues. The authors provide some strategies to ameliorate the problem of false peaks, though I am not sure how widely applicable they are (see below). I commend the authors for composing a very clear, well-written manuscript, with many great examples illustrating the problem.

Quantile normalization:

The authors normalized the fragment lengths between samples using a quantile normalization approach making the fragment length distribution the same across libraries. If I understood correctly, quantile normalization requires you to match the quantiles of the sample’s distribution to a standard distribution. How did the authors construct the standard size distribution - i.e. how was the blue line selected to be the standard to match all the other distributions to?

While the authors showed equalized fragment length distribution after quantile normalization, it is unclear to me how weighting the fragment lengths affects the intensity of ATAC-seq (or peak heights). I would like to see the variance of ATAC-peak heights between replicates before and after this normalization procedure.

This normalization approach between replicates makes sense to me but seems less justified between tissues (and to a certain extent genotypes). It makes the assumption that all the tissues should have the same size fragmentation distribution which implies all the tissues have the same global chromatin accessibility.

Accounting for structural variants:

Because the authors have high quality genomes for the strains, they were able to identify variable peak heights overlapping SVs. First, can the author provide some examples that are not due to TEs? Not saying TEs are not important - they absolutely are - I am just curious to see whether the same effects hold for other SVs like smaller indels or inversion. On this note, I would also like to see a breakdown of the fraction of false positives (ie. those removed after SV-correction) due to different types of SVs.

The need of high quality references tempers the wide-applicability of their accounting method - though I do not think it detracts from the quality of the paper. One way to also account for this would be to just use the DNA-seq from the strain as an input control and use the fold difference between ATAC-seq and DNA-input. This provides an enrichment value and is akin to ChIP-seq. And since the same mapping issues (due to TEs) or uneven mapping (due to duplications or indels) is expected to be in the input, you would expect some (but perhaps not all) of the issues mentioned in the manuscript to be mitigated without the need of a high quality reference or full accounting of all the SVs. This was previously done to look at reduced chromatin accessibility of TEs when TEs already have high coverage due to copy number (Wei, Chan, Bachtrog 2021).

Identifying causal SNPs:

It’s unclear to me how the SNPs in Figure 8 can be contributing to 100% of the ATAC variation. For example in the 8A Brain sample, there is one SNP but many different heights of the peak (also the SNP does not appear centered on the peak in the schematic, despite what the text describes in Line 437). Explaining 100% of the variation means that individuals in group1 have all the same SNP variant and the exact same quantitative phenotype (ie identical ATAC distribution in this case) and individuals in group2 all have the same SNP variant different from group 1 and have all have the same phenotype that is different from group 1. Based on the examples, I have no doubt that you are getting at causative variants (at least causative in the association-studies sense), but I am just not sure how you are estimating the effect sizes to get to >80% in some of these instances, as there are clearly more variation in the ATAC-seq distribution that can be explained by one SNP which i presume is binary.

Some other minor comments and issues for clarification:

When the authors say “Genotype” in their ANOVA tests, do they mean the strain? Or do they actually mean the genotype at/around the peaks?

Line 344, The authors mentioned there’s no evidence for hotspots for the variable ATAC peaks, but I would like to see a breakdown of the type of gene features they reside in (near TSS, CRE, exons, introns, 5’ and 3’ UTRs etc) - ideally separately for the genotype, tissue, and GxT categories.

Line 436, Do you mean Figure 8A?

Line 720: is the 5 in C+5 used as a pseudocount? Why was 5 chosen?

Line 745: is the "marker" variable the SNP genotype? Also what does the “1|” mean for the model?

For Figure 8, it would be great if the authors can incorporate the genotypes of the strains into the tracks. Currently there is no way of telling which Strain/SNP variant is associated with which ATAC-seq profile.

Reviewer #2: This is a very timely paper that asks whether there is genetic variation in chromatin accessibility. The question is addressed using a classical design, i.e. different genotypes of inbred strains are used to ATAC-Seq different tissues. The resulting data has 8 x 4 possible combinations of genotypes and tissues, and by replicating these treatment combinations, one can test for significance of genotype, tissue, and genotype by tissue interaction. This information is lacking in the literature in Drosophila because often only a single background/tissue is tested. I find the paper easy to read and the data set will be useful for the community. However, I do have several major concerns regarding the analyses and interpretations.

1. QC metrics should be reported for the ATAC-Seq bioinformatics, per sample/tissue/genotype, including at least, number of reads sequenced/mapped, uniquely mapped reads, number (percent) of reads within identified peaks, size (mean and sd) of peaks, etc. These pieces of information are important to understand the quality of the ATAC-Seq data and put some of the numbers in context. For example, the numbers of peaks identified across tissues may be a function of sequencing depth.

2. Figure 2: In the caption, better to call the all tissues set a union rather than consensus set. For depiction of the two peaks downstream of the 3'UTR of hairy in ovaries. Based on the text, the two peaks are separate with overlapping boundaries (black bars). Perhaps it's better to separate the black horizontal bars to indicate these are two different peaks? Is the y axis really fold enrichment? This seems like a coverage plot to me, where each base gets a sequencing depth.

3. There are other normalization methods for ATAC-Seq to deal with sample-to-sample variation in sequencing (e.g. TMM). The proposed method seems reasonable, but some additional evaluations are needed to help readers understand its performance and properties. For example, showing that the distribution of fragment size is identical after the normalization procedure (Figure 3) isn't enough to show that it works. Were PCR duplicates removed prior to the normalization? Could PCR duplicates lead to variation in the distribution of fragment sizes? I have one concern on the assumption behind the quantile normalization method. Say under the same peak near a gene, there are 2 150-bp fragments in Sample A, 20 150-bp fragments in Sample B but 38 300-bp fragments in Sample A and 20 in Sample B. This hypothetical data would lead to equal raw coverage for this peak in both samples, but would have drastically different coverage if they are quantile normalized. Can the authors provide some intuitive explanations why in such situations identical distribution of fragment sizes is the desired outcome?

4. The approach to account for SVs seems to be simply blacklisting SVs. If reference quality assemblies are available, would it be possible to map reads to their own references? Then do whole genome alignments to lift the coordinates over to a single reference? In other words, you still call peaks etc. on the main reference, but for quantitative assessment of peak coverages, map reads to their own genomes and lift coordinates back to the main reference. This will not only solve the SV problem but will presumably also improve mapping quality. Note that using this approach, SNPs will also be accounted for.

5. I do not think any of the analyses can claim causality for SNPs. In particular, variance partitioning using mixed models cannot determine causality. The true test is obviously experimentally editing bases, which would be beyond the scope of the current study. That said, I do believe that cis variants can cause polymorphism in chromatin structures, but it's a hypothesis that is best tested using other designs. Another caveat I want to point out is, if SNPs are not accounted for during mapping, there is a serious confounding issue because proximal SNPs may affect mapping of reads themselves. In other words, the causality may not be biological but mapping related artifact. I don't think this section is necessary, it actually weakens the paper.

Reviewer #3: In the work by Huynh et al., the authors sought to study the genetic variation in the chromatin state in Drosophila melanogaster. To achieve this goal, the authors sequenced and analyzed a number of ATAC-seq data from eight strains and four tissues. They performed a rigorous analysis of the project and had a number of interesting discoveries. The paper is well written; the dataset will be of great interest to many readers. The authors could have dug into some of the interesting observations further, but it is their decision. I only have a few minor comments.

1. One of the surprising findings is that 65.6% of the peaks are private to a single tissue. This is very interesting, but brain data largely affects the result. I’d recommend the authors also report the tissue-specific peaks by tissues. For example, eye and wing discs have a much lower percentage of tissue-specific peaks.

2. The authors found that most peaks have fold enrichment of less than 5. What does that mean? Is it because most peaks have residual open chromatin regions nearby, or because most of the peaks only exist in a small number of cell types within a tissue (e.g., cell type-specific peaks with tissue-level background noise)? I understand the authors’ data may not give a definitive answer. Nevertheless, it would be interesting to check it.

3. It is unclear how many of the peaks located in exon/UTR/intergenic regions in each tissue are shared. For example, it is likely that promoter/TSS regions of universally expressed genes are shared by other intergenic peaks are more tissue specific. Maybe I missed it somewhere in the manuscript, but it would be interesting for readers to know.

4. The authors provided a nice example: without considering TE-related indels, the comparisons can be screwed. Is it a rare case? In total, how many such TE insertions are found in the eight strains?

5. Line 424-428, I am not sure if I understand “a total of 22 SNPs annotated as having a high functional impact”. Why would SNPs with protein sequence or structures (missense, premature start codon, or splice variant) could fully explain 100% of the peak height? It is theoretically possible to have linked regulatory SNPs in the nearby regions, but the authors do not seem to suggest so.

6. It is unclear to me, in the method, how the authors accounted for structural variants. I understand the authors have de novo reference genomes for each strain. With indels, the coordinates will be different at all locations on the genome. How the authors used de novo genomes and provide information for standard genomes?

**Have all data underlying the figures and results presented in the manuscript been provided?**

Reviewer #1: Yes

Reviewer #2: Yes

Reviewer #3: Yes

PLOS authors have the option to publish the peer review history of their article (what does this mean?). If published, this will include your full peer review and any attached files.

Reviewer #1: **Yes: **Kevin Wei

Reviewer #2: No

Reviewer #3: No

---

## [Decision Letter · Decision Letter 1]

27 Mar 2023

Dear Dr Long,

Thank you very much for submitting your Research Article entitled 'Genetic Variation in Chromatin State Across Multiple Tissues in Drosophila melanogaster' to PLOS Genetics.

We acknowledge and appreciate that your revision of this manuscript was very thorough. The manuscript was fully evaluated at the editorial level and by two independent peer reviewers.  Reviewer #1 has a few remaining comments/questions, which we expect will be straightforward to address and/or reply to.  After this the paper will be acceptable for publication.  

We therefore ask you to modify the manuscript according to the review recommendations. Your revisions should address the specific points made by each reviewer.

Yours sincerely,

Kelly A. Dyer

Academic Editor

PLOS Genetics

Kirsten Bomblies

Section Editor

PLOS Genetics

Reviewer's Responses to Questions

**Comments to the Authors:**

Reviewer #1: The authors addressed most of my concerns and questions, but I have some lingering issues that the authors should clarify.

Quantile normalization of fragment lengths:

I had asked the authors to demonstrate how their quantile normalization affects peak calling and peak intensity. The authors replied that this is not useful, and mentioned they provided supplementary table 4 with mapping stats and supplementary figure 8. The point of my comment was that the authors do not show how the quantile normalization strategy affects downstream inference of chromatin accessibility (i.e. the process of calling peaks and quantifying intensity/accessibility). Does this manipulation/normalization make the ATAC-seq signals (peaks and peak intensity) more robust and less variable? How does peak intensity (or even coverage over peaks) change using the quantile normalization procedure vs. just normalization by library size. Are these values more or less variable between replicates in this normalization scheme compared to more rudimentary methods? The question I am getting at is whether quantile normalization produces more robust quantitative and qualitative characterization of accessibility and chromatin states. As far as I can tell, the authors merely demonstrated that their quantile normalization procedure makes the fragment size distribution more robust - but the more meaningful question is whether accessibility measures are more robust.

Regarding whether tissues should have the same fragment size distribution, I agree mostly with the authors that most tissues are probably quite similar, therefore justifying the use of fragment size distribution (quantile) normalization, at least in their case. But there are instances where this assumption is problematic - early embryo development and zygotic genome activation when the chromatin landscape undergoes major changes. The chromatin environment during this time is of particular interest to a lot of researchers, and subject to MANY ATAC-seq studies. I highly recommend the authors caveat their approach and mention when the assumption (of similar fragment size distribution) may not be met.

Example of SVs causing false peaks.

Upon my request of including a SV not related to TEs, the authors provide one example of a 1970bp deletion causing a false peak. I am a bit confused by this example since as the authors mentioned this leads to the erroneous inference of closed chromatin. But in the ATAC-seq tracks they show, in the uncorrected track, the deletion is associated with an increased ATAC-seq signal (and a downstream peak). Also, I would highly recommend that the authors include at least one of such indel-associated issues in the supp, instead of only supplying for reviewer eyes.

Reviewer #2: The authors have adequately addressed my concerns. I have no further comments.

**Have all data underlying the figures and results presented in the manuscript been provided?**

Reviewer #1: Yes

Reviewer #2: Yes

PLOS authors have the option to publish the peer review history of their article (what does this mean?). If published, this will include your full peer review and any attached files.

Reviewer #1: **Yes: **Kevin Wei

Reviewer #2: No

---

## [Editor Report · Decision Letter 2]

20 Apr 2023

Dear Dr Long,

We are pleased to inform you that your manuscript entitled "Genetic Variation in Chromatin State Across Multiple Tissues in Drosophila melanogaster" has been editorially accepted for publication in PLOS Genetics. Congratulations!

Yours sincerely,

Kelly A. Dyer

Academic Editor

PLOS Genetics

Kirsten Bomblies

Section Editor

PLOS Genetics

Comments from the reviewers (if applicable):

**Data Deposition**

http://datadryad.org/submit?journalID=pgenetics&manu=PGENETICS-D-22-01092R2

**Press Queries**

---

## [Editor Report · Acceptance letter]

3 May 2023

PGENETICS-D-22-01092R2 

Genetic Variation in Chromatin State Across Multiple Tissues in Drosophila melanogaster 

Dear Dr Long, 

We are pleased to inform you that your manuscript entitled "Genetic Variation in Chromatin State Across Multiple Tissues in Drosophila melanogaster" has been formally accepted for publication in PLOS Genetics! Your manuscript is now with our production department and you will be notified of the publication date in due course.

With kind regards,

Zsofi Zombor

PLOS Genetics

On behalf of:
